# A Porcine Model for the Development and Testing of Preoperative Skin Preparations

**DOI:** 10.3390/microorganisms10050837

**Published:** 2022-04-19

**Authors:** Hannah R. Duffy, Rose W. Godfrey, Dustin L. Williams, Nicholas N. Ashton

**Affiliations:** 1Department of Orthopaedics, University of Utah, Salt Lake City, UT 84112, USA; hannah.duffy@utah.edu (H.R.D.); u1115789@utah.edu (R.W.G.); dustin.williams@utah.edu (D.L.W.); 2Department of Biomedical Engineering, University of Utah, Salt Lake City, UT 84112, USA; 3Department of Pathology, University of Utah, Salt Lake City, UT 84112, USA; 4Department of Physical Medicine and Rehabilitation, Uniformed Services University of the Health Sciences, Bethesda, MD 20814, USA

**Keywords:** biofilm, surgical site infection, antiseptic, preoperative skin preparation, porcine model, povidone iodine, chlorhexidine gluconate

## Abstract

Clinical preoperative skin preparations (PSPs) do not eradicate skin flora dwelling in the deepest dermal regions. Survivors constitute a persistent infection risk. In search of solutions, we created a porcine model intended for PSP developmental testing. This model employed microbiological techniques sensitive to the deep-dwelling microbial flora as these microorganisms are frequently overlooked when using institutionally-entrenched testing methodologies. Clinical gold-standard PSPs were assessed. Ten Yorkshire pigs were divided into two groups: prepared with either povidone iodine (PVP-I) or chlorhexidine gluconate (CHG) PSP. Bioburdens were calculated on square, 4 cm by 4 cm, full-thickness skin samples homogenized in neutralizing media. Endogenous bioburden of porcine skin (3.3 log_10_ CFU/cm^2^) was consistent with natural flora numbers in dry human skin. On-label PSP scrub kits with PVP-I (*n* = 39) or CHG (*n* = 40) failed the 2–3 log_10_-reduction criteria established for PSPs by the Food and Drug Administration (FDA), resulting in a 1.46 log_10_ and 0.58 log_10_ reduction, respectively. Porcine dermal microbiota mirrored that of humans, displaying abundant staphylococcal species. Likewise, histological sections showed similarity in hair follicle depths and sebaceous glands (3.2 ± 0.7 mm). These shared characteristics and the considerable fraction of bacteria which survived clinical PSPs make this model useful for developmental work.

## 1. Introduction

Preoperative skin preparation (PSP) is the cleansing and disinfection of intact skin before an invasive procedure; it constitutes a clinical standard as it significantly reduces the prospect of surgical site infection (SSI) [1]. As part of a broad antiseptic technique, PSPs have greatly reduced SSIs since their inception in the mid-19th century [1,2,3]. Prior to the implementation of antisepsis, fatalities from infection in the most prestigious European hospitals hovered around 40 to 50%. Today, ~2–5% of clean procedures result in SSIs with associated mortality rates of ~1.5% [4]. Early antiseptic efforts focused on cross-patient contamination, comprising hand-cleaning techniques for medical personnel and instruments, but quickly grew to encompass the patient’s endogenous skin flora [5,6]. With the introduction of Joseph Lister’s own nascent protocols, his self-reported post-operative survival rates climbed from 55% to 85% [2]. Diverse antiseptic agents and protocols have since been devised for PSP; among the earliest included the use of chlorinated lime, carbolic acid, and boric acid [1,7,8,9]. Today, medical practitioners have largely settled on alternating scrubs of alcohol and solutions with either chlorhexidine gluconate (CHG) or the iodophor povidone iodine (PVP-I); these chemical agents strike a balance between rapid microbicidal activity and low toxicity to patients’ tissues [10].

As these commonplace topical agents were developed throughout the mid-1900s, one of the pioneers of open-heart surgery poignantly reflected on the contemporary beliefs of the day regarding PSP procedures: “that by the use of surface agents perfection in the sterilization of skin (as in everything else) is impossible and must remain, as it always has been, a goal and a standard for relative comparison [11]”. Indeed, our current understanding of PSPs and their antimicrobial efficacy remains scarcely changed—a fact which has been recently reemphasized in a thorough 2021 meta-analysis of PVP-I and CHG PSPs which states, “Human skin cannot be sterilized even by intense disinfection [10]”.

The patient’s endogenous microflora is the single greatest contributor to unwanted surgical site bioburden, exceeding exogenous contributions from instruments, room air, clinical personnel, etc. [12]. Abundant direct and indirect evidence support these assertions. First, skin flora contaminants are ubiquitous in the open surgical site, surgical instruments, surgeon’s gloves, and newly placed orthopedic hardware [13,14,15,16,17]. Moreover, skin flora vary by anatomical location; these variations are mirrored by the site-specific pathogen profile of associated SSIs [12]. The most conspicuous example, *Cutibacterium acnes* (previously *Propionibacterium acnes*)—the typical causative agent of acne vulgaris—occupies pilosebaceous follicles, which are in the highest density in the skin of the scalp, shoulders, back, upper chest, breasts, and face [18,19]. Percutaneous procedures in these sites are beset by *C. acnes* as the most abundant pathogen identified in SSIs, surpassing even staphylococcal species [20,21,22,23,24]. In other anatomical locations, staphylococcal species are most abundant as they occupy the ubiquitous sweat glands and hair follicles [19,25,26]. As such, staphylococcal infections are omnipresent in nearly all types of transdermal procedures [27]. This direct correlation between endogenous dermal flora and SSI culprits cannot be overlooked.

The Food and Drug Administration’s (FDA) regulatory requirements for PSPs reflect these known technical limitations in antimicrobial efficacy. Their requirements, in essence, acknowledge the inability of PSPs to create a truly sterile surgical field. PSPs are regulated through the FDA’s Tentative Final Monograph for Healthcare Antiseptics, which utilizes methodology from the ASTM standard E1173-15 [28]. Briefly, human efficacy testing of PSPs is performed on two distinct anatomical locations: the moist inguinal region with higher bioburdens and the dry abdominal region with lower bioburdens (10^5−7.5^ and 10^2.5−3.5^ CFU/cm^2^, respectively) [29]. The FDA criteria for PSPs requires an initial 3 and 2 log_10_ reduction at 10 min for the two respective sites (many surgeons now commonly prep skin for less than 1 min). Effectively, surgical sites treated with FDA-cleared PSPs allow upwards of 10^2–4^ CFU/cm^2^ on the skin surface as the surgical access incision is made in the operating room. The presence of these low-level survivors in deeper skin regions is conversely documented in studies utilizing the FDA methodology [29,30].

The FDA’s acceptable level of residual bioburden is well below the minimum infectious dose thresholds of 10^5^ CFU/g or CFU/mL established for opportunistic pathogens, such as those in skin microflora. The significance of the 10^5^ CFU/g threshold has been observed in numerous animal and human studies with origins in work from the mid-1900s [31,32,33,34,35,36]. The clinical relevance of the 10^5^ CFU/g infection threshold has been deemed “the 10^5^ guideline” or “the 10^5^ rule” [37,38]. Yet, there is an undeniable caveat to the apparent 10^5^ rule—implanted biomaterials potentiate infection from very low-level contamination, greatly reducing the minimum infectious bacterial dose [12]. Work performed in 1957 showed that a sterile suture, one of the few biomaterials of the day, decreased the minimum pus-forming dose of *Streptococcus pyogenes* from 10^6^ to 10^2^ CFU/g in human volunteers [39,40]. This 1957 observation of artificial materials as foci of infection was presciently perched on the cusp of a biomaterials revolution; artificial biomaterials have since become commonplace in clinical medicine for their restorative capacity [41]. Examples include vascular grafts (1952), heart valves (1953), prosthetic hip joints (1958), prosthetic knee joints (1968), and implantable pacemakers (1959), among the earliest of the ever-growing list of biomaterial implants [41].

The etiology of device-related infections, and how artificial surfaces potentiate infection, is now well understood [42,43,44,45]. Opportunistic pathogens preferentially form sessile communities on implant biomaterials and devitalized tissues. These multicellular communities are called biofilms [42]. The biofilm strategy protects bacteria from host immunity and antimicrobial chemotherapeutic interventions. Among other evasive strategies, the biofilm deranges phagocytic clearance by host leukocytes, which are unable to engulf small bacterial aggregates if they are larger than their own diameter (10–12 µm) [46,47,48,49,50]. Moreover, the biofilm harbors quiescent phenotypes which display recalcitrance to antimicrobial chemotherapeutics because they have downregulated metabolic drug targets [51]. This subpopulation of biofilm constituents has been referred to as persister cells because they can outlast the most aggressive courses of systemic antibiotics [52,53]. The bacteria lodged in the deeper dermal regions reside as more-stationary biofilm phenotypes, with persister subpopulations tolerant of prophylactic antibiotics [54,55]. Biofilms underpinning device-related infections constitute a significant risk to the patient; biofilm formation on implant materials undermines the restorative and life-saving outcomes intended by surgery.

PSP procedures have changed very little over the last half-century. Yet, there is an ever-present risk from the low-level inocula known to survive clinical PSP because of an insatiable proclivity towards infection-potentiating biomaterials in modern surgical sites. The primary motivation for this study was the development of a suitable porcine model with which experimental PSPs can be developed for human use. The relevance of this porcine model and associated methodology was assessed against two gold-standard clinical PSPs used on label: alcohol-based wet skin scrub trays with PVP-I or CHG.

## 2. Materials and Methods

### 2.1. Supplies, Instruments, and Reagents

Dey-Engley (D/E) neutralizing broth and tryptic soy broth (TSB) were purchased from MilliporeSigma (Burlington, MA, USA), Petri dishes and agar from Fisher Scientific (Hampton, NH, USA), and brain heart infusion (BHI) broth from Research Products International (Mt Prospect, IL, USA). Alcohol-based Wet Skin Scrub Tray with CHG (CHG preoperative skin preparation kit, 4% solution) from Medline (Northfield, IL, USA) and Alcohol-based Wet Premium Skin Scrub Trays (PVP-I) preoperative skin preparation kit, 7.5% scrub and 10% paint) from BD (Franklin Lakes, NJ, USA), as shown in Figure 1. Magic Bullet blenders, cups, and blades from Homeland Housewares LLC (Los Angeles, CA, USA). Columbia blood agar plates and AnaeroGen Anaerobic Gas Generator from Hardy Diagnostics (Santa Maria, CA, USA). Due to supply chain issues stemming from COVID-19, AnaeroGen Anaerobic Gas Generator packets were unavailable for a portion of the work in this study. This necessitated the use of commercial hand warmers in combination with an anaerobic indicator for assurance; this is a well-established workaround [51]. Bacteria isolate identification was outsourced to Nelson Labs (Salt Lake City, UT, USA). Surgical tools, histological processing equipment/materials, and other miscellaneous supplies were provided by the Bone and Biofilm Research Lab (Salt Lake City, UT, USA).

### 2.2. Tissue Sample Collection

Ten Yorkshire pigs underwent surgery with approval and oversight of the Institutional Animal Care and Use Committee (IACUC) at the University of Utah. Either a PVP-I or CHG PSP was applied using aseptic technique by a trained individual approved by IACUC. Hair was clipped then clean shaved along the entire back of each pig. Ten pigs were randomly divided into two groups (*n* = 5/group). The first group (Pigs 1–5) received an alcohol-based PVP-I surgical scrub (Figure 1A). The second group (Pigs 6–10) received an alcohol-based CHG surgical scrub (Figure 1B). Each PSP was applied according to the manufacturer’s instructions and current clinical standard of care; PSPs were applied on surgical site in concentric circles followed by 70% isopropyl alcohol (each 3x) and allowed to dry 5 min before the tissues were surgically removed.

In total, 8 square, 4 cm by 4 cm, full-thickness skin samples were collected from each animal: a total of 39 PVP-I-treated samples (one PVP-I sample was discarded due to procedural errors, leaving only 39 samples instead of 40) and 40 CHG-treated samples. Skin samples were harvested using aseptic technique: 3 cm away horizontally from the spine and at least 2 cm apart (Figure 2). Samples were placed individually into 50 mL conical tubes filled with 35 mL of sterile D/E broth to neutralize the antiseptic agent (Figure 3). The conical tubes were then placed and transported in a chilled cooler to mitigate bacterial replication. Samples were homogenized and the bioburden was quantified by serial dilution plating (see section below for details).

After 28 days, each pig was euthanized and underwent necropsy. This study was run using available animals from a separate primary study. As such, control tissues which did not receive PSP were collected immediately after the animals were sacrificed (28 days after the CHG and PVP-I samples were collected) to completely comply with the approved animal use protocols of the primary study. To collect control tissue samples, the skin was wiped at least 3x with sterile water and gauze to remove visible detritus. Four additional 4 cm by 4 cm square skin samples were collected at necropsy using sterile surgical supplies to obtain a bioburden baseline in the pig skin. Post-necropsy sample collection sites did not overlap with the boundaries of the original sample collection sites to reduce confounding factors of the skin that had been treated with a presurgical scrub (Figure 2A).

### 2.3. Tissue Processing and Bioburden Quantification

The 8 oz blender cups and blades were cold sterilized by exposing all internal parts to 200 proof ethanol for >40 min (>20 min on each side). The cups and blades were air dried in a closed biosafety cabinet until all ethanol evaporated. Alternatively, sterile deionized water was added to the blender cup and blended to remove/dilute any residual ethanol. Each skin sample was transferred from a 50 mL conical tube to the sterile blender cup and homogenized for 5 min in approximately 1-min intervals to allow the mixture to cool. During blending, blender cups were manually agitated to ensure thorough homogenization of the mixture.

Skin samples were transferred back into their respective conical tubes, which were then vortexed for 60 s, sonicated for 10 min, and vortexed again for 10 s. Two hundred µL of the sample mixture were plated in duplicate and spread with a sterile loop on Columbia blood agar for a 0-dilution plate. Bioburden was quantified using a 10-fold serial dilutions series: 100 µL of the sample mixture was serially diluted in 900 µL of PBS carried through to 10^−4^. Moreover, 50 µL of each dilution was plated on Columbia blood agar, dispersed in 5 × 10 µL spots, to quantify the CFU/cm^2^ of each skin sample. Each sample was plated in duplicate, which were divided and incubated under aerobic and anaerobic conditions. Aerobic culture plates were incubated for 48 h at 37 °C in a jacketed incubator. Anaerobic culture plates were incubated at 37 °C for 72 h in an anaerobic jar with an anaerobic gas generator and oxygen indicator. Colony counts were used to represent the bioburden in units of CFU/cm^2^.

### 2.4. Processing Controls

This process was also performed in the absence of skin samples to confirm that the processing technique was void of contamination. Blenders were sterilized according to the same procedure in Section 2.3. Sterile D/E broth was blended, transferred to 50 mL conical tubes, vortexed, and sonicated, as previously mentioned. CFU counts were performed and reported in the same way as the tissue samples.

### 2.5. Bacterial Characterization

After incubation, bacterial colonies were classified according to morphology and color. A representative colony of each distinct morphology from the aerobically grown and anaerobically grown plates of each of the three study groups (Control, PVP-I, and CHG) was isolated with a sterile loop, streaked onto Columbia blood agar, and incubated at 37 °C aerobically or anaerobically according to its original growth conditions. Each bacterial isolate was cataloged and frozen (−80 °C) in 1 mL of sterile 70% BHI broth with 30% glycerol.

Cryo-preserved samples were cultured on TSB agar and incubated at 37 °C for 24–48 h in their respective aerobic or anaerobic environments. Each selected isolate was characterized using the Gram-staining method with safranin as the counterstain. The shape of each bacterial isolate was determined using transmission light microscopy. The Gram stain, shape, and morphology were documented for each isolate. Pie charts were created including every isolate from the six different test conditions: Control Aerobic, Control Anaerobic, PVP-I Aerobic, PVP-I Anaerobic, CHG Aerobic, and CHG Anaerobic.

Cross-comparison of all of the organisms using Gram-staining, organism shape, colony morphology, and colony color was used to reduce all of the bacterial isolates down to the unique bacterial species across all study groups. The unique species that survived the PSPs were genotyped using the MicroSeq microbial identification system with MicroSEQ ID 16S rDNA 500 Microbial Library, v2019 database comparison (Thermo Fisher). Genotypic identification was performed by Nelson Labs (Salt Lake City, UT, USA) on 16 unique isolates. Four isolates common only to the control skin samples were likewise genotyped.

### 2.6. Histology

During the final necropsy of pig 5 and pig 10, a selection of full-thickness skin samples (4 cm by 4 cm) was collected for histological analysis. The samples were fixed in 10% neutral buffered formalin with 3 × 24 h changes and then dehydrated in increasing concentrations of ethanol and infiltrated with xylene in a Tissue-Tek 6 AI vacuum infiltration processor (Sakura). They were then placed under vacuum in a solution of 80% methyl methacrylate, 20% dibutyl phthalate, and 2.5 g/L Perkadox 16 for 24 h at 4 °C. Samples were then transferred onto a bed of pre-polymerized methyl methacrylate/dibutyl phthalate/Perkadox 16 solution and polymerized for 24 h (using UV light) in a 2.5 cm layer of that same solution. The final poly-methyl-methacrylate (PMMA)-embedded sample was then trimmed down on a bandsaw to remove excess PMMA and ground and polished to an optical finish before being glued to a slide using Tecknovit 7210 VLC (Exakt).

Slides were placed on an IsoMet 1000 Precision Cutter creating uniformed thin (>1 mm) sections and then ground by hand to a final thickness of ~75 µm. The ground slides were stained with Sanderson’s Rapid Bone Stain (SRBS) for 1 min 30 s at 55 °C and viewed under a light microscope (Nikon E600, Nikon Inc., Melville, NY, USA) equipped with associated image capturing software (Camera Control Pro, Nikon, Melville, NY, USA). Digital images were captured throughout the epidermis, dermis, and subcutaneous tissue and post-processed using Microsoft Research Image Composite Editor (MRICE) to mosaic/stitch, creating an overhead view.

### 2.7. Statistics

Multiple locations on each pig’s back were used, which introduced data clustering. Therefore, we analyzed the data using a mixed effects linear regression (also called a multilevel model), which accounted for locations nested within the pig [56]. In our model, experimental condition was a fixed effect and pig was a random effect. Given the small sample size, the models were specified to use a significance test based on a t statistic, rather than the default z statistic, and fitted using Stata-17 statistical software (StataCorp LLC, College Stata, TX, USA).

The following comparisons were made: PVP-I Aerobic vs. Control Aerobic, PVP-I Anaerobic vs. Control Anaerobic, CHG Anaerobic vs. Control Aerobic, CHG Anaerobic vs. Control Anaerobic, PVP-I Aerobic vs. CHG Aerobic, PVP-I Anaerobic vs. CHG Anaerobic, PVP-I Aerobic vs. PVP-I Anaerobic, CHG Aerobic vs. CHG Anaerobic, Control Aerobic vs. Control Anaerobic, PVP-I Aerobic vs. Process Control, and CHG Aerobic vs. Process Control.

### 2.8. Calculating Approximate Penetration Depth

Diffusion coefficients of PVP-I, CHG, and chlorhexidine (the active ingredient in CHG) were approximated from the Wile and Chang correlation using water as the solvent assuming 37 °C [57]. The diffusion coefficients for iodine (I^-^, the active ingredient in PVP-I) and isopropyl alcohol (IPA) were acquired from literature references [58,59].

Using ImageJ Software and two histological images of hair follicles, seven representative measurements were taken from the bottom of the stratum corneum to the surface of the porcine skin from histological sections. Seven additional measurements were taken along the hair follicle, measured between the bottom-most tip of the hair follicle to the point at which the hair passed through the skin.

A table was generated reporting the approximate diffusion times for PVP-I, I^-^, CHG, chlorhexidine, and IPA to the bottom of the stratum corneum and the hair follicle. Assuming unidirectional flow and random walk, we calculated the approximate penetration time along the hairs and skin using Equation (1), an approximation from Fick’s second law, where diffusion time (*t*) is approximated by the square of the diffusion distance (*x*) divided by the diffusion coefficient (*D*) [60]:(1)$$t\approx \frac{x^{2}}{2D}$$

## 3. Results

### 3.1. Bioburden Quantification

The average dermal bioburden of native pig flora across the samples of the 10 study animals was 3.3 log_10_ CFU/cm^2^. The average bioburden for those used in PVP-I analysis (Pigs 1–5) was 3.4 ± 0.2 log_10_ CFU/cm^2^ (aerobic culture) and 3.2 ± 0.3 log_10_ CFU/cm^2^ (anaerobic culture). For the pigs used in the CHG analysis (Pigs 6–10), the average bioburden was 3.2 ± 0.2 log_10_ CFU/cm^2^ (aerobic culture) and 3.0 ± 0.2 log_10_ CFU/cm^2^ (anaerobic culture). The bioburden of native pig flora was highly varied, ranging from 15 CFU/cm^2^ (Pig 8 Anaerobic Control, tissue sample 4) to 3.3 × 10^4^ CFU/cm^2^ (Pig 2 Anaerobic Control, tissue sample 2). Considerable variability was also observed within the same animal across different areas of the back (Figure 4).

There was similar variability in the residual bioburden for samples prepared with PVP-I or CHG PSPs. For PVP-I-treated samples, the residual bioburden levels ranged from 0 to 3.8 × 10^3^ CFU/cm^2^; 17 out of 78 samples had 0 CFU/cm^2^ (combined aerobic and anaerobic cultures). The average bacterial levels per animal (*n* = 8 tissue samples/pig) are shown in Figure 4 with the quantity of PVP-I-persisting bioburden in orange (Figure 4A); the average bioburden persisting through PVP-I (log_10_ transform) was 1.7 ± 0.3 CFU/cm^2^ (aerobic culture) and 1.6 ± 0.3 CFU/cm^2^ (anaerobic cultures). For CHG, 3 out of 80 samples had 0 CFU/cm^2^ with a high end of 1.7 × 10^4^ CFU/cm^2^ (combined aerobic and anaerobic cultures). The average bacterial levels per pig (*n* = 8 tissue samples/pig) are shown in Figure 4 with the quantity of CHG-persisting bioburden in blue (Figure 4B). Using the CHG preparation, the resulting bioburden was 2.8 ± 0.3 log_10_ CFU/cm^2^ (aerobic culture) and 2.6 ± 0.4 log_10_ CFU/cm^2^ (anaerobic culture).

The log_10_ reduction from the natural flora (control) was calculated for the PVP-I- (*n* = 39) and CHG (*n* = 40)-treated dermal samples (Figure 5A). The PVP-I preparation reduced the bioburden by 1.5 ± 0.5 log_10_ and 1.4 ± 0.5 log_10_ units for aerobic and anaerobic culture conditions, respectively. Likewise, the CHG kits reduced the aerobic and anaerobic bioburden by 0.6 ± 0.5 log_10_ and 0.6 ± 0.7 log_10_ units, respectively. Contamination in the process controls (Figure 5A) was minimal, with an average of 11 ± 1 CFU/cm^2^ (averaged across a hypothetical 4 cm by 4 cm sample area) or 0.4 ± 0.2 log_10_ units. Most of these controls (72%, *n* = 18) resulted in clean plates (0 CFU/cm^2^).

### 3.2. Statistics

The most relevant *p*-values for this study are listed in Figure 5A. In both aerobic and anaerobic culture conditions, comparing the PVP-I-scrubbed tissue with the controls resulted in a *p* value of <0.001. Comparing CHG with the corresponding controls yielded *p*-values of 0.002 and 0.012 for aerobic and anaerobic culture conditions, respectively. Comparing PVP-I and CHG resulted in *p* values of 0.039 and 0.084 for aerobic and anerobic culture conditions, respectively. The process control data were also compared to the anerobic PVP-I condition (the condition with the lowest bioburden), resulting in a *p* value of 0.003. None of the comparisons between the aerobic and anaerobic within the same animal group were significant.

### 3.3. Bacterial Identification

Multiple types of bacteria were identified from each study animal. Exactly 186 bacterial isolates were distributed as follows: 61 from the aerobic controls pooled from pigs 1–10, 24 from anaerobic controls pooled from pigs 1–10, 29 from the aerobic PVP-I from pigs 1–5, 19 from the anaerobic PVP-I from pigs 1–5, 31 from the aerobic CHG from pigs 6–10, and 22 from the anaerobic CHG from pigs 6–10. The above numbers are represented by the surface area of the pie charts in Figure 6, and thereby serve as the denominators for data presentation and discussion. The majority of the microorganisms were facultative anaerobes growing in both culture conditions. The porcine microflora comprises a more diverse subset of Gram-positive microbes by a wide margin over Gram-negative organisms. Regardless of the group, there was much less diversity of Gram-negative isolates. In-fact Gram Negative isolates were only identified in plates from the aerobically grown control. Additionally, rod-shaped bacteria were more likely to grow aerobically as they were absent in anaerobically grown cultures—indicating that porcine dermal rod-shaped bacteria are strict aerobes. One isolate could not be positively identified in the anaerobically grown positive control group.

The bacterial species which survived both PSPs were: *Staphylococcus epidermidis, Staphylococcus chromogenes, Staphylococcus haemolyticus, Staphylococcus saprophyticus, Bacillus thuringiensis, Corynebacterium glutamicum*, and *Corynebacterium variabile*. One species was identified which only survived in the CHG PSP: *Bacillus pumilus*. These surviving bacteria were also ubiquitous in the control groups, as identified through similarities in Gram staining, bacterium shape, colony morphology, and colony color. Four species of bacteria were exceptionally common but only in the flora of the control groups: *Bacillus aerophilus, Bacillus cereus, Micrococcus luteus, and Staphylococcus xylosus*. In short, porcine skin treated with PSPs displayed a reduction in the diversity of bacteria, although the exact extent is not fully known, as several unique isolates in the control samples were not genotyped due to budgetary constraints.

### 3.4. Histology and Diffusion

The histological images for a representative porcine skin sample, oriented in plane with the hair follicles, are included in Figure 7. The thin stratum corneum (asterisk), the deepest part of the hair follicles, and the sebaceous glands were all easily visible using polarized light and transmission light microscopy in unison with the Sanderson’s rapid bone stain (Figure 7). Anatomical measurements were performed across four different sections of skin. The follicle depths were (*n* = 7): 2.40, 2.56, 3.94, 2.48, 4.03, 2.87, and 3.86 mm. The stratum corneum depths were (*n* = 7): 0.028, 0.031, 0.051, 0.047, 0.104, 0.088, 0.043, and 0.056 mm. The average and respective standard deviation of these lists are reported in the first column of Table 1.

The approximate diffusion coefficients for PVP-I, I^-^, CHG, chlorhexidine, and IPA are shown in the top row of Table 1. Below these coefficients are the respective penetration depths and standard deviations into the follicle and stratum corneum. The ratio between the diffusion time into the follicle and stratum corneum was about 3 × 10^3^. Diffusion times were correlated to molecular weight with I^-^ diffusing fastest, followed by IPA, PVP-I, Chlorohexidine, and, finally, CHG. Evaporation of these molecules was not considered in this analysis.

## 4. Discussion

The majority of endogenous skin flora are associated with the superficial stratum corneum [10,61]. These are more successfully managed with the mechanical and chemical means of clinical PSPs [10,61]. Their eradication accounts for the considerable microbial reduction observed for clinical PSPs in FDA filings (>2 log_10_). The fraction of surviving microbes (FSM) measured after PSP principally comprises the deeper-dwelling less-transient skin flora along the sebaceous glands, sweat glands, and hair follicles—a fact which was established as early as 1945 through histological observations that “Bacteria are situated so deep in the hair follicles and sebaceous glands that generally used antiseptics do not penetrate sufficiently to reach the organisms located in the deeper parts of these structures” [62].

The standard required by the FDA for assessing PSP efficacy in live human volunteers (ASTM E1173-15) relies on a methodology which is, by necessity, non-destructive to the skin, but insensitive to the organisms lodged deeper in the dermal tissues [28]. Briefly, a sterile containment cylinder is placed over the skin site which is then filled with neutralizing broth while the site is gently massaged with a sterile rubber scraper to dislodge residual transient microbes on the skin surface. The dislodged bioburden in the broth is quantified using standard serial dilution plating. Thus, the bioburden numbers reported using FDA methodology are likely underestimates, devoid of information regarding the microbes lodged in deeper skin layers. To avoid this blind spot in the porcine model we developed herein, full-thickness skin samples were homogenized in neutralizing broth to quantify the true FSM. This provides a more accurate understanding of the problem. This change from FDA methodology complicates cross-data comparisons in exchange for better representation of the true magnitude of infection-causing microbes deep in the skin. Notably, this procedure is destructive to the tissue and only suitable for animal work and human applications where the skin is surgically discarded for other reasons.

The baseline bioburden measured in this porcine model (3.34 log_10_ CFU/cm^2^) is consistent with that reported for drier abdominal human sites (3.36 log_10_ CFU/cm^2^) [29]. As pigs do not have sweat glands, this comparison is likely more realistic than a comparison with the moist inguinal human sites with much higher baseline microbial counts (6.16 log_10_ CFU/cm^2^) [29]. The reduction in bioburden from baseline in this porcine model did not exceed the 2 log_10_ FDA requirement for dry abdominal sites for either the PVP-I or CHG PSPs: 1.46 ± 0.52 and 0.58 ± 0.50 log_10_, respectively for aerobically cultured samples. Similar log_10_ reductions were observed for anaerobically cultured samples: 1.43 ± 0.52 and 0.55 ± 0.67 log_10_ respectively. Because the measured bioburden in the tissue samples is similar to that of drier human sites, the inability of the PSPs to achieve the necessary log_10_ reduction could not be due to a significantly higher starting bioburden in the porcine skin. A higher FSM might stem from the more thorough sampling procedure or might result from variations in skin anatomy that more completely shroud microbes from chemical antisepsis and mechanical removal. We view the lackluster performance of clinical PSPs as an essential component of a robust model for developing and testing experimental PSPs; it means that there is a considerable signal for detecting improvements in developmental PSPs over clinical gold-standard PSPs.

This porcine model exhibits similar microflora characteristics to human skin. During the process of Gram staining, Gram-positive cocci, consistent with the *Staphylococcaceae* family, predominated in more than half of our samples. In humans, *S**taphylococcus* species are the culprits of a high percentage of SSIs. In the selection of samples that were sent for identification, 5 of the 12 species identified belonged to the *Staphylococcous* genus, with 4 of the 16 total samples identified as *Staphylococcus epidermidis*. Additionally, *Corynebacteriaceae*, *Micrococciaceae*, and *Bacillaceae* bacterial families are also commonly found in human skin, all of which include facultative anaerobic species that can grow in both aerobic and anaerobic conditions. This may explain why differences between aerobic and anaerobic reductions were not statistically significant [63,64,65]. We hypothesize that the same types of bacteria grew aerobically as well as anaerobically. There was a greater diversity in bacterial species in native flora compared with the skin treated with PSPs. The reduction in biodiversity might point towards a select group of microbes specializing and surviving in the niches of the follicle and sebaceous glands, which are outside of the diffusive influence of the PSP antimicrobials.

Histological analysis of porcine skin (Figure 7) was performed to enable interpretation of the PSP microbial efficacy reported here and for comparison to human skin anatomy. The depth of the porcine follicles measured here (3.2 ± 0.7 mm) is in good agreement with the follicular depths reported across a wide range of human dermal sites: the trunk (2–4.5 mm), arms (2.5–4 mm), legs (2.5–4 mm), beard (2–4 mm), and breasts (3–4.5 mm) [66]. The porcine sebaceous glands, as in human skin, have a similar depth as their companion follicle. For consideration, human sweat glands, known to harbor staphylococcal species, plumb comparable depths as the follicles and sebaceous glands: ~4.2 mm. The measured depths for porcine stratum corneum (56 ± 29 µm) were, on average, larger than those reported for humans (6.2–19.4 µm) but with considerable overlap [67].

Clinical PSPs are expeditious, usually occurring just several minutes before the surgical access incision. Antimicrobials only reach the deep dermal regions through passive Fickian diffusion [10]. Under these diffusive conditions, the time it takes for an antimicrobial to reach its target is proportional to the square of distance from the skin surface as shown by Equation (1). Diffusion times are exceptionally small over the short sub-millimeter distances of the stratum corneum (1–4.8 s, Table 1), but at the depths of the average follicle (3.2 mm) estimated diffusion times soar, ranging from 44 to 218 min for the various antimicrobials in the PSPs tested here. These diffusion times (Table 1) are low-end approximations. The diffusion coefficients used in the Table 1 calculations assume a liquid–liquid system with water as the diffusive medium. Diffusion is significantly hindered by both mechanical barriers and electrostatic interactions between the molecule and charged tissue components; the mobility of these antimicrobials is likely hindered beyond what is accounted for in the diffusion coefficients [68]. Despite these simplified calculations, it is clear that antimicrobial compounds do not reach sufficient concentrations within the short timeframe of PSPs to eradicate the bacteria dwelling in the deepest dermal regions. These survivors constitute a considerable infection risk especially in procedures using implanted biomaterials known to potentiate infection from these low-level inocula.

## 5. Conclusions

The sterile surgical field, even after an on-label application of a PSP, is a myth, reflecting wishful thinking or a laudable goal, at best. Although PSPs substantially reduce bioburden, they do not kill or dislodge bacteria dwelling in the deep dermal layers. The microbial survivors constitute a significant SSI risk as implanted biomaterials, which are ubiquitous in the modern surgical site, potentiate infection from low-level contamination. Our porcine model for testing PSPs mirrors, to a substantial degree, the human condition. Importantly, we developed microbiological methodologies that are sensitive to bacteria dwelling in the deeper dermal regions. The fraction of bacteria surviving clinical gold-standard PSPs in this model is large; this provides a substantial signal for improvement when testing experimental PSPs. We therefore conclude that this model will be useful for developmental research towards improved PSP technologies and intend to use it accordingly.

## Figures and Tables

**Figure 1 microorganisms-10-00837-f001:**
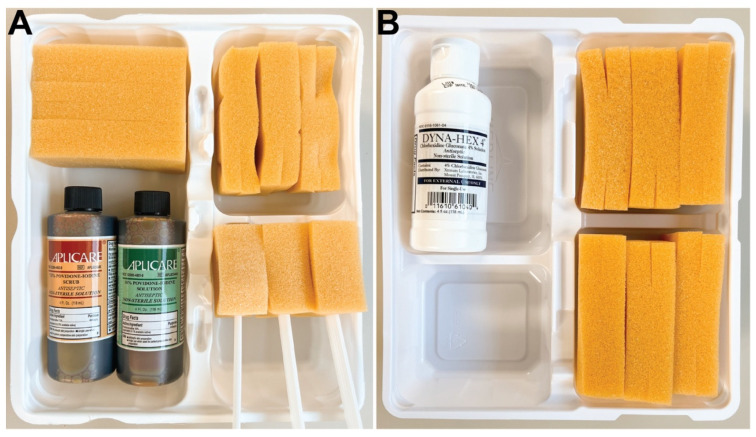
PVP-I and CHG presurgical scrub trays used for this work. Swabs included in each kit were not used during the preparation: (**A**) Alcohol-based Wet Premium Skin Scrub Tray with PVP-I (PSP kit, 7.5% scrub and 10% paint solution) from BD was used for pigs 1–5; (**B**) Alcohol-based Wet Skin Scrub Tray with CHG (PSP kit, 4% CHG solution) from Medline was used for pigs 6–10.

**Figure 2 microorganisms-10-00837-f002:**
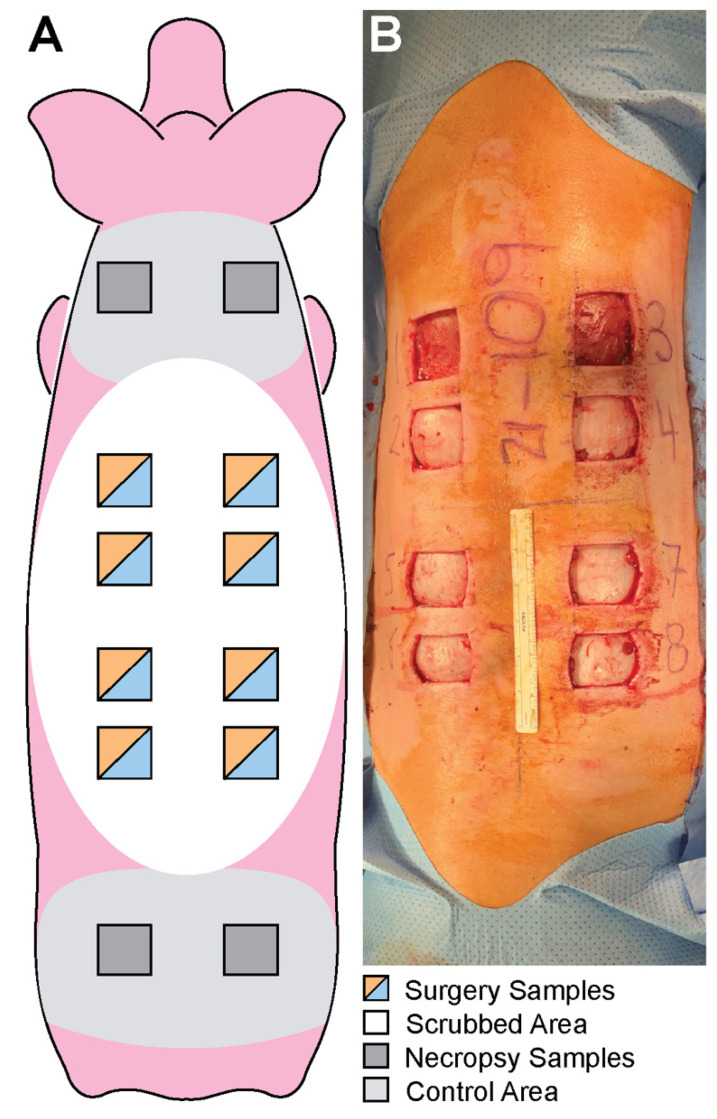
(**A**) Diagrams of the anatomical location of the skin preparation, control area, and the respective skin samples collected. (**B**) Dorsal view of surgery with the skin samples removed after PSP. Samples were cut 3 cm away horizontally from the spine and at least 2 cm apart.

**Figure 3 microorganisms-10-00837-f003:**
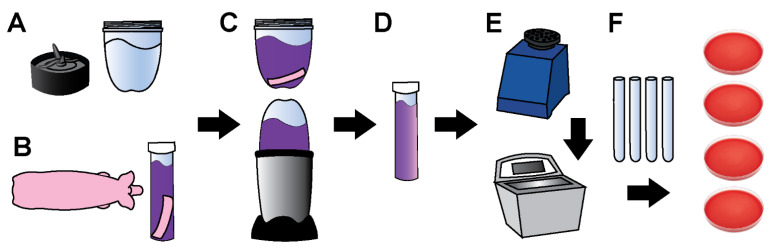
Sequential method for processing skin tissues collected and processed using PSP model using full-thickness samples: (**A**) Cold sterilization of magic bullet blenders with 200 proof ethanol; (**B**) Square skin samples (4 cm by 4 cm) placed in 35 mL of D/E neutralizing broth; (**C**) Sample blended for approximately 5 min; (**D**) Sample transferred back to the original conical tube; (**E**) Sample vortexed for 1 min, sonicated for 10 min, vortexed for 10 s; (**F**) 10-fold serial dilution performed in PBS. The 10^0^ dilution and 10^−1^–10^−4^ dilutions were plated in duplicate on separate Columbia blood agar and incubated in aerobic or anaerobic conditions.

**Figure 4 microorganisms-10-00837-f004:**
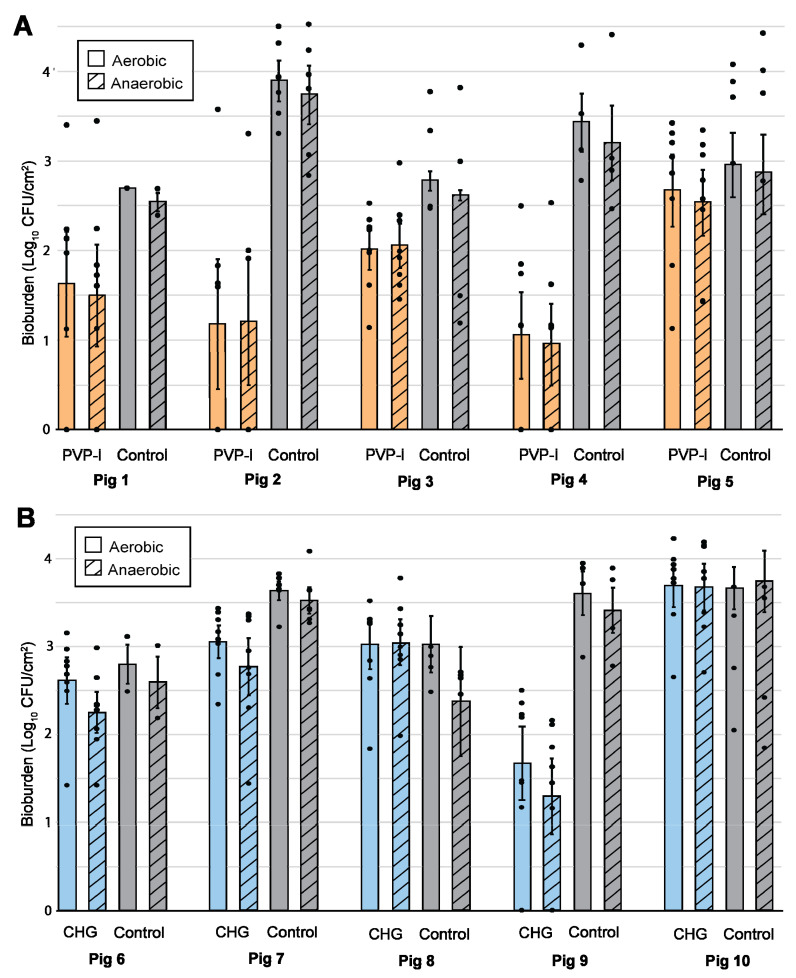
The log_10_-transformed bioburden of the tissue samples collected following PSP compared to the control samples. Each bar is a representation of the average per pig with the points representing the individual tissue samples. Each PSP bar has 8 points from eight tissue samples while each control bar has 2–6 points respective of the number of samples taken. A solid bar indicates aerobic culture conditions while hatching indicates anaerobic culture conditions. (**A**) Pigs 1–5 received a PVP-I scrub (Orange). (**B**) Pigs 6–10 received a CHG scrub (Blue). The error bar shows ± S.D.

**Figure 5 microorganisms-10-00837-f005:**
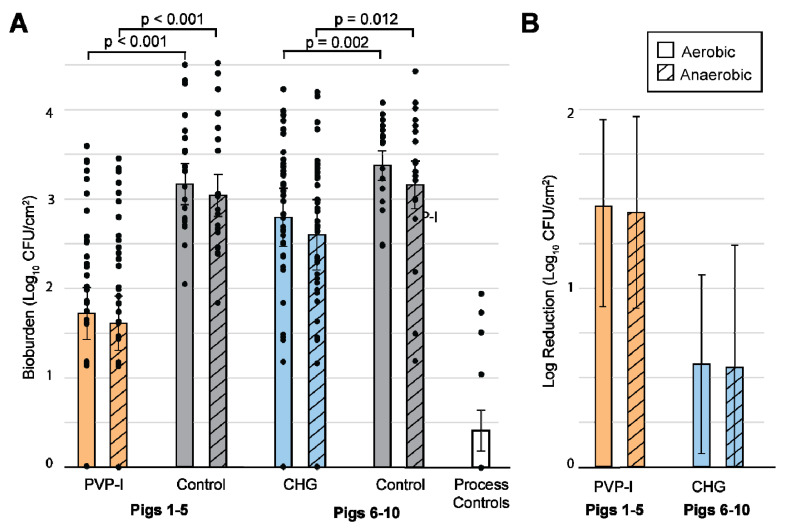
Bioburden for each PSP and the corresponding controls. Each bar is the average of each test condition. The points shown along each bar represent all individual tissue samples pooled together. Error bars are the standard error from the mixed regression model to represent clustered data. A solid bar indicates aerobic condition while hatching indicates anaerobic condition. (**A**) shows bioburden in log_10_ CFU/cm^2^ with an axis of 0–4, while (**B**) shows log reduction with an axis of 0–2. In each chart, the pig numbers corresponding to the respective treatments are shown. A selection of *p* values is listed above the brackets indicating the data that were compared. In both aerobic and anaerobic conditions, comparing PVP-I-scrubbed tissue with controls resulted in a *p* value of <0.001. Comparing CHG with the corresponding controls yielded *p* values of 0.002 and 0.012 for aerobic and anaerobic conditions, respectively.

**Figure 6 microorganisms-10-00837-f006:**
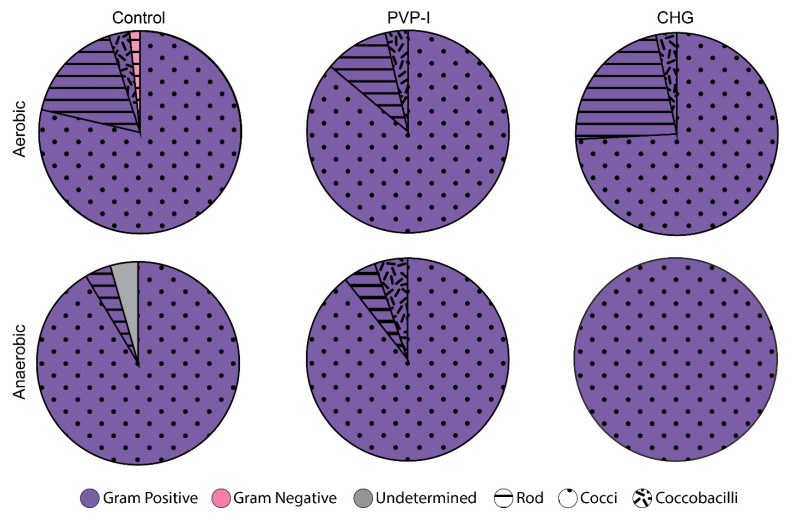
Distribution of every bacteria isolate from the three study groups characterized by Gram-stain analysis and morphology. Exactly 186 bacterial isolates were distributed as follows: 61 from the aerobic controls pooled from pigs 1–10, 24 from anaerobic controls pooled from pigs 1–10, 29 from the aerobic PVP-I from pigs 1–5, 19 from the anaerobic PVP-I from pigs 1–5, 31 from the aerobic CHG from pigs 6–10, and 22 from the anaerobic CHG from pigs 6–10. Gram-positive bacteria are shown with purple while Gram-negative bacteria are shown in pink. The shape (Rod, Cocci, or Coccobacilli) is indicated by the textures. The area of each pie chart represents the respective number of isolates identified in each group.

**Figure 7 microorganisms-10-00837-f007:**
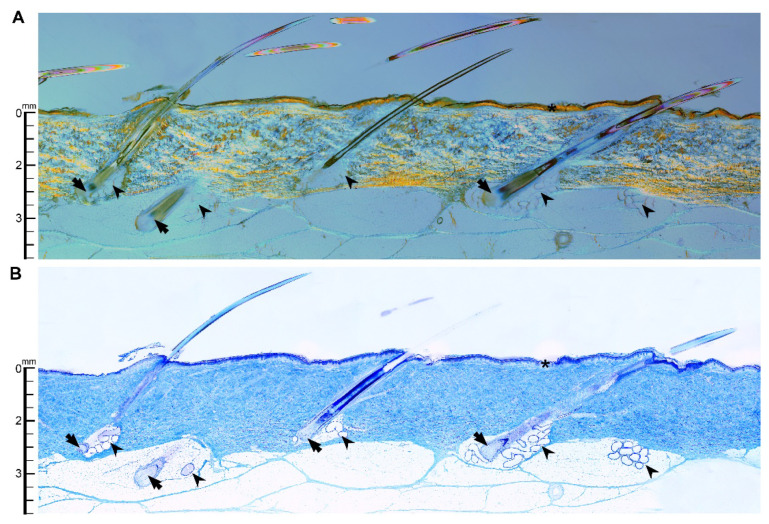
Representative longitudinal section (~75 µm thick) of PMMA-embedded skin using polarized light (**A**) and transmission light (**B**) microscopy with Sanderson’s rapid bone stain. Notable features are shown: the thin stratum corneum (asterisk), the deepest part of the hair follicles (arrow), and the sebaceous glands (arrowhead). The scale bars along the left are oriented to illustrate the approximate depth (in mm) of tissue features from the stratum corneum.

**Table 1 microorganisms-10-00837-t001:** Approximate diffusion coefficients and corresponding diffusion times are listed as a comparative reference.

	Depth (mm)	PVP-I	I^-^	CHG	Chlorhexidine	IPA
Diffusion Coefficient		6 × 10^−10^ m^2^/s [56]	2 × 10^−9^ m^2^/s [57]	4 × 10^−10^ m^2^/s [56]	5 × 10^−10^ m^2^/s [56]	1 × 10^−9^ m^2^/s [58]
Follicle	3.2 ± 0.7	145 ± 67 min	44 ± 20 min	218 ± 101 min	175 ± 80 min	87 ± 40 min
Stratum Corneum	0.056 ± 0.029	3.2 ± 3.2 s	1.0 ± 1.0 s	4.8 ± 4.8 s	3.9 ± 3.9 s	1.9 ± 1.9 s

## Data Availability

Data are available in this manuscript.

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
