# Peer review of "A Porcine Model for the Development and Testing of Preoperative Skin Preparations"

_microorganisms, 2022, doi:10.3390/microorganisms10050837_

Round 1

Reviewer 1 Report

A article entitled, "A porcine model for the development and testing of preoperative skin preparation ", written by Duffy, et al. describes a model that might be useful for evaluation of bacterial evacuation under conventional skin preparation for the surgery.  The concern that deep-seated bacteria in the hair follicles and sebaceous glands might become a source of opportunistic infections or form biofilms on implant biomaterials is well-established, but an in vivo model is long overdue for improving sterilization procedures to suppress such deep-seated microbial flora. Therefore, this study is well-chosen and welcome. The experiments are reasonably organized with proper controls and the writing was nicely done. However, the data are poorly presented and render the conclusions overly obscure and unsuitable for publication as they stand.  In particular, please consult with a more experienced person to make Figures 4-6 legible. Further recommendations are as follows:

  • Figure 4 would be better with a scatter plot than a bar graph. Averages with error bars are appropriate here, but it makes no sense to provide an error bar for each pig unless you see a systemic bias in your results (in which case that is an issue you must address).
  • The degree of inhibition is likely to be a more useful number than the absolute CFU, so again, the dataset should be reduced further before it is presented.
  • In general, P-values should be indicated in the graphs themselves and not in a separate table.
  • The population distribution (and the shifts therein) among the flora under the various treatments are likely to be of greater interest to clinicians than the simple fact of survival. Figure 6 as it stands seems to do an infelicitous job of making this point.
  • While both the PVP-1 and CHG preparations, and especially the CHG group, failed to keep the CFU below 100 in Figure 5, the bacterial distribution in Figure 6 could shed some new light on both treatments. It seems that PVP-1 and CHG reduce the diversity of bacteria, especially aerobic bacteria. Unfortunately, the discussion of this point is lacking.
  • To better correlate bacterial distribution between the two treatments, I suggest that you either indicate the total number of bacteria in the control, PVP-1 and CHG groups (described in beginning of 3.2. Bacterial identification) in the Figure 6 pie figures, or make a new graph to reflect the inhibitory effects of bacterial distribution under each treatment.   
  • Table 2 is meaningless and lends no support to your conclusion. For example, epidermidis should be the most common species on the skin, but it is not found in the control samples. I suggest rearranging Table 2 (example below) by listing groups (control, PVP-1 and CHG) in the first column and growth conditions (aerobic/)anaerobic) in the second and the third column:

Groups

Aerobic (found in samples)

Anaerobic (found in smaples)

Control

xxx

xxxx

PVP-1

S.epidermidis (4)

xxx

CHG

S.epidermidis (4)

xxxx

  • In this study only one time point is reported (5 min). Were other periods considered?  I wonder if longer or shorter periods would have an effect on the quantity and quality of bacterial flora.  For example, the diffusion times given in Table 3 suggest that a much longer period of treatment might be in order.

Author Response

Response to Reviewer 1 Comments

Response: The authors are appreciative of the reviewers careful reading of the manuscipt and the insightful and constrictive critiques graciously provided. To assure we incorprated correctly the requested changes we consulted with the University of Utah’s Biostatistical Support for help with data analysis and statistics.     

Point 1: Figure 4 would be better with a scatter plot than a bar graph. Averages with error bars are appropriate here, but it makes no sense to provide an error bar for each pig unless you see a systemic bias in your results (in which case that is an issue you must address).

Response 1: As suggested we added the raw data values as scatter plots to figure 4. We chose to overlay the scattered data with the original color-coordinated bar grphas to help guide the reader with quickly visible population statistics: mean, and standerd deviation.

A systemic bias refers to sampling methods which consistently undersetimate or overestimate the true value. We have no eveidecne of a systemic bias in this study as all of the samples were collected in an identical manner. We believe it still makes sense to provide the individual date and error bars for each pig because of considerable variation in endogenous bioburden from animal to aniaml. As part of presenting these reuslts, to aid in replication of the study, It is important to note that individual animals did not have the same residual bioburden to start with. For researchers wanting to replicate our results, a comparison of the individual bioburdens per pig would be more useful than the cumulative data as future researchers will likely also use pigs whose bioburden is also distributed differently.

A desire for consolidated data was hinted at by the revier in this comment. This is still included in figure 5 which as been modified to account for clustering of the data; the controls were separated and plotted along with the treatment groups shared by the same pigs. Likewise, the raw data was ovrelayed as as scatter plot on these figures. The  

Point 2: The degree of inhibition is likely to be a more useful number than the absolute CFU, so again, the dataset should be reduced further before it is presented.

Response 2: We added a pannel B to figure 5 showing the requested analysis with properly propagated errors. We chose to still include pannel A because raw bioburden levels are also important for interpritation of infection risk.  For example, the minimim infectious dose for healthy tissues is ~105 CFU/ gram and lower if an artificial material is present ~102-3. The methods and results were changed to reflect the new analysis.      

Point 3: In general, P-values should be indicated in the graphs themselves and not in a separate table.

Response 3: As requested, we included the most important p values in Figure 5A; those indicating significant difference between each PSP and its respective control. We did not include all possible p vlaues in the figure because we wanted the chart to be easy to read and understand. These additional p values for each comparison from the methods section is presented in the body of the results. We consulted the department of Biostatistics to review our statistical methods. Per their suggestion, data clustering was incorporated and thus, standard deviations, standard errors, and p values shifted accordingly. We are confident that this is a much more statitically robust representation of our data. 

Point 4: The population distribution (and the shifts therein) among the flora under the various treatments are likely to be of greater interest to clinicians than the simple fact of survival. Figure 6 as it stands seems to do an infelicitous job of making this point.

Response 4: Our previous explanation of our methods of creating Figure 6 seemed to be unclear to our reviewers. We have reviewed our confusing language and adjusted accordingly for clarity. The Pi charts in figure 6 contain the gram-stain analysis on all of the identitified isolates from each of the six groups, they do not contain MicroSEW 16s RNA idetification. Outputs of the gram-stain analysis include: if an orgnisim is Gram positive or negative; and the respective shape of the bacterium (Rod, Cocci, Cocobacili, spirochete, spirilla, vibrio, etc..). We choose to include the presentation of this data, as it represents the Gram stain analys performed on every isolate identified in the study. This study did not have the budget to get 16S rRNA identification on all study isolates; this at least, privides some identification in the absence of more costly analyis. Only 16 of the hundreds of isolates identified in this study were sent for MicroSEQ 16S rRNA analysis.  All of the study isolates are cryo-preserved, as we work on obtaining funding for future analysis.   

Point 5: While both the PVP-1 and CHG preparations, and especially the CHG group, failed to keep the CFU below 100 in Figure 5, the bacterial distribution in Figure 6 could shed some new light on both treatments. It seems that PVP-1 and CHG reduce the diversity of bacteria, especially aerobic bacteria. Unfortunately, the discussion of this point is lacking. 

Response 5: Figure 6 shows the distribution of all bacterial isolates characterized by microscopic morphology and Gram stain analysis. Analyisis of 186 isolates comprises figure 6.

 Because of budget constrants, only 16 isolates were genotyped using the MicroSeq microbial identitication system. MicroSeq identification was focused primaril on those unique species that survived the surgical skin preparations. All of the surviving bacterial species were characterized. To perform this analysis, Gram stain information along with colony morphology of streak plate isolates were used to remove redundant samples from those which underwent costly genotyping. All of the species which survived the PSPs were also ubiqitous in the control samples. There ware Four common species of bacteria in the controls which weren’t identified in the PSP (CHG nor PVP-I) samples. These were also sequenced. In short, porcine skin treated with PSPs display a reduction in diversity of bacteria alt-hough the exact extent is not fully known as several unique isolates in the controls samples were not genotyped due to budgetary constraints. We have changed the confusing wording in these sections and ammended our methods to make the methods clear for the reader. We added this insightful discussion of bacterial diversity to the discussion section in lines as suggested 578-582.    

Point 6: To better correlate bacterial distribution between the two treatments, I suggest that you either indicate the total number of bacteria in the control, PVP-1 and CHG groups (described in beginning of 3.2. Bacterial identification) in the Figure 6 pie figures, or make a new graph to reflect the inhibitory effects of bacterial distribution under each treatment.  

Response 6: See reaponse 5 for a more in depth discussion of this. We understand this was confusing an updated our methods and results. For the results regarding the exact genotypes of bacteria in each sample as requested see updates in lines 392-402 of the updated manuscript.

Point 7: Table 2 is meaningless and lends no support to your conclusion. For example, epidermidis should be the most common species on the skin, but it is not found in the control samples. I suggest rearranging Table 2 (example below) by listing groups (control, PVP-1 and CHG) in the first column and growth conditions (aerobic/anaerobic) in the second and the third column:

Groups

Aerobic (found in samples)

Anaerobic (found in smaples)

Control

xxx

xxxx

PVP-1

S.epidermidis (4)

xxx

CHG

S.epidermidis (4)

xxxx

Response 7: Table 2 was removed because of the confusion with its interpritation. This data was replaced with the wording in the results section (See lines 392-402)

Point 8: In this study only one time point is reported (5 min). Were other periods considered?  I wonder if longer or shorter periods would have an effect on the quantity and quality of bacterial flora.  For example, the diffusion times given in Table 3 suggest that a much longer period of treatment might be in order.

Response 8: This is a great point, one which would be worth persuing in the future using this model. This is not within the scope of this particular study. In this study, we used clinical PSPs on label. The on label use of these kits constitutes a 5 min presurgical drying time. We would certanly anticipate longer exposeru times corresponding with better performance of the PCPs but to what extent is unknown. We would like to point out, most surgical prcedures are rushed and the 5 min preparation is only an ideal and likely represents a high-end estimate for performance.

Reviewer 2 Report

This study created a porcine model intended for clinical preoperative skin preparations (PSP) developmental testing. 

It is surprised that on-label PSP scrub kits with PVP-I or CHG failed the 2-3 log10-reduction criteria by the FDA, resulting in a 1.62 log10 and 0.54 log10 reduction only. How do author explain this? Thanks.

Is it due to the porcine living environment is more dirty than human, has more bacterial flora reside in deeper dermal regions or due to improper skin preparation? Need more detailed information of the PSP preparation. 

Author Response

Response to Reviewer 2 Comments

Response: We appreceiate the thoughtufl review and used the suggestions to improve the manuscript.

Point 1: This study created a porcine model intended for clinical preoperative skin preparations (PSP) developmental testing. It is surprised that on-label PSP scrub kits with PVP-I or CHG failed the 2-3 log10-reduction criteria by the FDA, resulting in a 1.62 log10 and 0.54 log10 reduction only. How do author explain this? Thanks. Is it due to the porcine living environment is more dirty than human, has more bacterial flora reside in deeper dermal regions or due to improper skin preparation? Need more detailed information of the PSP preparation. 

Response 1: There was already considerable discussion regarding this topic in the discussion section. We have suggested that the numbers reported in human trials are ‘likely underestimates, devoid of information regarding the microbes lodged in deeper skin layers.’ because the sample methodology required by the FDA is insensitive to the bacteria in the deeper dermal regions: Lines 538-552. We have added extra claification to the discussion and comparision of porcine dermal bioburden and human dermal bioburden: Lines 553-567. In short, pigs aren’t more dirty than humans. , the inability of the PSPs to achieve the necessary log10 reduction could not be due to a significantly higher starting bioburden in the porcine skin. It might be due to differences in anatomical features, hair folicles and sebacious glands, but we also discuss how their dimensions are similar. Our best guess is that our technique of blending tissues raher than scrubbing the surface, more perfectly accounts for the bacteria dwelling in the deeper dermal regions.

Point 2: Need more detailed information of the PSP preparation. 

Response 2: As suggested we updated the methods section to include more granular deatil of the PSP procedures performed in the OR on these animals. We are appreceiative of the feedback .

Round 2

Reviewer 1 Report

No more comments for your revised MS.